

# Prediction of yield and quality in medicinal plant *Ligusticum chuanxiong* Hort. using uncrewed aerial vehicle multispectral measurement

Yun-Fan Li[1,2], Chen Wu[1,2], Hong-Mei Jia[1,2], Xi Chen[1,2], Jin-Niu Xing[1,2], Wei-Ping Gao[1,2] and Zhu-Yun Yan[1,2]

[1] State Key Laboratory of Southwestern Chinese Medicine Resources, Chengdu University of Traditional Chinese Medicine, Chengdu, China

[2] School of Pharmacy/School of Modern Chinese Medicine Industry, Chengdu University of Traditonal Chinese Medicine, Chengdu, China

Corresponding author
Zhu-Yun Yan,
yanzhuyun@cdutcm.edu.cn

## ABSTRACT

Accurate predicting the yield and quality of medicinal materials before harvest can effectively guide post-harvest process, including processing and storage, thereby ensuring the final quality of medicinal materials. Currently, traditional experimental methods for yield and quality estimation are inadequate to offer reliable guidance for harvesting and processing of medicinal plan. Uncrewed aerial vehicle (UAV) multispectral can quickly and accurately estimate the yield and quality of field crops. Based on the UAV multispectral data of *Ligusticum chuanxiong* Hort. obtained about half a month before and near harvest, this study predicted the rhizome yield and the content of active components such as ferulic acid, Z-ligustilide and senkyunolide A. Additionally, the quality discriminant models of chuanxiong rhizoma were constructed according to the ferulic acid content index stipulated in Pharmacopoeia of the People's Republic of China (2020). The results performed on the independent validation set show that the best prediction effects of fresh weight and dry weight of rhizome were NRMSE = 23.76%, MAPE = 14.75% and NRMSE = 34.65%, MAPE = 21.73%, respectively. And the best predictive effects of ferulic acid, Z-ligustilide and senkyunolide A were as follows: NRMSE = 13.35%, MAPE = 10.25%; NRMSE = 34.35%, MAPE = 23.40%; and NRMSE = 45.26%, MAPE = 25.48%. Furthermore, the quality discriminant models XGBoost and AdaBoost had effective performances (Accuracy = 0.7083, AUC = 0.7214). These results suggest that UAV multispectral can be effectively employed to predict both yield and quality before harvest, thereby guiding the harvest and processing of *L. chuanxiong*.

## INTRODUCTION

Cultivation of medicinal plants in China has a long history and is still thriving (*Song et al., 2024*). In 2020, the total planting area of Chinese medicinal plants is about 55, 596 km² (*Wang et al., 2022*). By 2025, there will be about 1,667 km² cultivation bases of Chinese medicinal plants in China (*Wang et al., 2023a*). The quality and economic value of Chinese
medicinal materials are directly related, and the quality of Chinese medicinal materials is not only affected by cultivation methods and collection time, but also by processing, transportation and storage processes in the production area (*Yang et al., 2024*). Therefore, it is necessary to prepare the human and material resources according to the yield and quality of medicine plants to complete those processes timely and efficiently for maximizing the utilization of Chinese medicine resources. How to quickly and accurately estimate the yield and quality of Chinese medicine materials before harvesting medicinal plants is an important issue in the current Chinese medicine planting and production industry.

Uncrewed aerial vehicles (UAVs), a low-altitude independent sensing technology, have become one of the most popular tools in precision agriculture production management applications due to its flexible and convenient operation and quick access to crop-related data in specific regions (*Alckmin et al., 2022*). The growth of multiple crops can be monitored by high-throughput phenotypic data collected by UAV (*Karunathilake et al., 2023*), such as corn (*Shahhosseini et al., 2019*), wheat (*Upreti et al., 2019*; *Wang et al., 2021*), rice (*Cen et al., 2019*), sugarcane (*Shendryk et al., 2020*), soybean (*Maimaitijiang et al., 2017*), potato (*Jasim et al., 2020*), sweet potato (*Tedesco et al., 2021*) and other crops. The multispectral sensor can obtain the spectral reflectance data of more than two bands on the crop surface without contact, and can calculate the Normalized Difference Vegetation Index (NDVI) and other vegetation indexes (VIs) related to crop growth status and biomass according to the data (*Radočaj et al., 2023*; *Wang et al., 2023a*). At present, the machine learning algorithm model combined with VIs derived from multispectral independent sensing images can accurately monitor crop growth, including biomass and quality prediction (*Zhang et al., 2022a*; *Zhang et al., 2022b*). For example, the use of UAV multispectral data can not only construct regression prediction models of table beet root weight (*Chancia et al., 2021*) and cassava underground biomass (*Selvaraj et al., 2020*), but also effectively evaluate the soluble solids content of grape (*Lyu et al., 2023*) and industrial poppy thebaine alkaloid content (*Iqbal, Lucieer & Barry, 2020*). However, UAV multispectral is mostly used for crop yield and quality prediction and rarely for medicinal plants.

The Chinese medicinal material, chuanxiong rhizoma, is the dried rhizome of *L. chuanxiong*, which is widely used in the treatment of cardiovascular and cerebrovascular diseases (*Chen et al., 2022*; *Li et al., 2022*). The quality of chuanxiong rhizoma is affected by the timely fresh processing and the storage conditions (*Yu et al., 2021*; *Yan et al., 2019*). Predicting the yield and quality of *L. chuanxiong* before harvesting is important to rationally manage the post-harvest handling process. In the previous study, our team used UAV multispectral to construct models for detecting water and nutrient deficiencies of *L. chuanxiong* (*Li et al., 2023*). Therefore, UAV multispectral can be a potential technical platform for estimating the yield and quality of *L. chuanxiong*. The main purpose of this study are divided into two points: (1) The regression model for predicting the yield of *L. chuanxiong*, the content of three active components in chuanxiong rhizoma, and the quality discriminant model of chuanxiong rhizoma based on FA content were obtained, and the accuracy and fitting degree of the model were tested; (2) the obtained models is used to predict the independent validation set in the same period and evaluate the

universality of the models. It is expected to provide a feasible method for predicting the yield and quality of *L. chuanxiong* before harvest.

## MATERIALS & METHODS

### Study area

The study areas, Shiyang Town in Dujiangyan City (30.85°N, 103.66°E) and Aoping Town in Pengzhou City (31.10°N, 104.00°E), were in the Chengdu Plain (Fig. 1), which lies in western China and has a subtropical humid monsoon climate. The altitude of Aoping Town is about 581.7 m, the average annual temperature is 15.7 °C, and the average annual rainfall is 924.7 m$^3$, while the altitude of Shiyang Town is about 698.5 m, the average annual temperature is 15.2 °C, and the average annual rainfall is 1,177.5 m$^3$ (*Yin et al., 2012*). In addition, Shiyang Town is the traditional genuine producing area of chuanxiong rhizoma, while Aoping Town has the largest planting area and yield (*Li et al., 2012*; *Li et al., 2024*).

### UAV image collection

The UAV platform was the DJI Phantom 4 Multispectral four-wing UAV (DJI, Shenzhen, China), which was equipped with a multispectral lens (FC6360), including a color sensor for visible light imaging and five monochromatic sensors for multispectral imaging (B: 450 ± 16 nm, G: 560 ± 16 nm, R: 650 ± 16 nm, RE: 730 ± 16 nm, NIR: 850 ± 26 nm), the resolution power of each sensor was 2.08 million pixels. The top of the UAV has a light intensity sensor that could be used for radiometric correction of later data. The flight geographical coordinates were determined by real time kinematic (RTK) GPS system with horizontal and vertical kinematic errors less than one cm and 1.5 cm respectively.

About half a month before the harvesting period (April 27, D1) and near the harvesting period (May 8, D2) in 2023, respectively, the UAV images of *L. chuanxiong* fields in Shiyang Town were obtained. In addition, the UAV images in Aoping Town was collected near the harvesting period (May 3, D3). The images were all obtained from 11:00 to 14:00 on the same day, there was no cloud cover to block the sun, and the low wind speed did not affect the flight. During the flight, the recording angle to the ground was 90° and the flight speed was 6 m/s. The flight altitude was 50 m, with a ground sampling distance (GSD) of 2.65 cm pixel$^{-1}$. Images were acquired with 85% forward overlap and 60% lateral overlap. Prior to each flight, the image of two radiation calibration panels with reflectance values of 10% and 90% were obtained at an altitude of 1.5 m.

### Measurement of *L. chuanxiong* rhizome fresh weight and dry weight

*L. chuanxiong* was mined in Aoping Town (May 4, 2023) and Shiyang Town (May 8, 2023). Each sample consisted of three rhizomes of *L. chuanxiong* in the center of a square sampling area of one m$^2$. A total of 125 samples from six different planting fields in Shiyang Town were used as the training and test sets. In addition, a total of 24 samples were collected from eight different planting fields of in Aoping Town as the independent validation set. After harvesting, the above-ground parts, roots and impurities such as soil attached to the rhizomes were removed and fresh weight (FW) was measured. After the rhizomes were brought to the laboratory and dried at 105 °C for 30 min, then dried at 50 °C until the

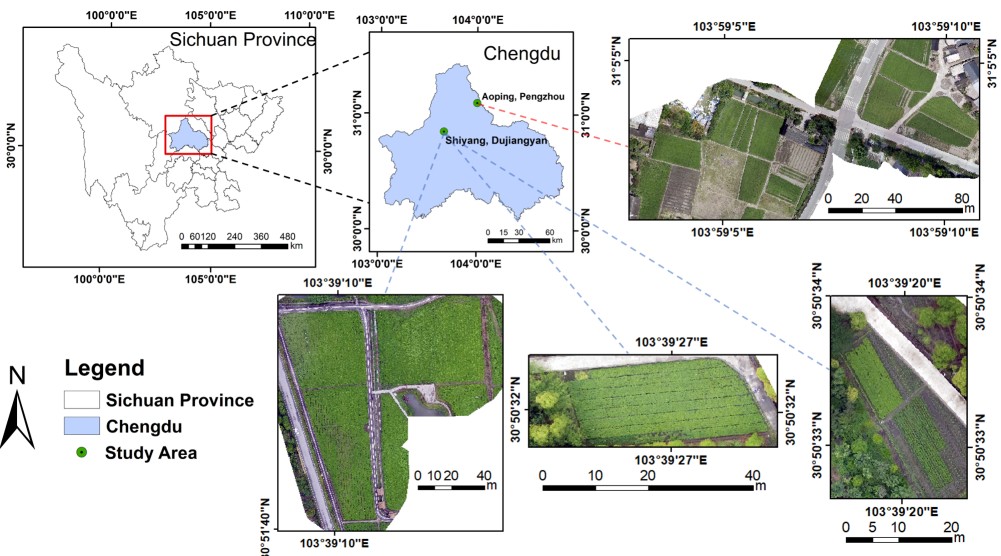

**Figure 1  Location of the study area.**

weight remained unchanged, the dry weight (DW) was determined. The FW and DW of each sample were the average of the FW and DW of the three rhizomes in the sample.

## Measurement of active component contents
### Preparation of standard substance solutions

Ferulic acid (FA), Z-ligustilide (Z-L) and senkyunolide A (SA) standard substances were dissolved in methanol, respectively, as the standard solution mother solutions. Then, they were diluted into standard substance solutions with concentrations of 31.92 $\mu$g mL$^{-1}$, 275.00 $\mu$g mL$^{-1}$ and 160.00 $\mu$g mL$^{-1}$ for quantitative analysis, respectively.

### Preparation of sample solutions

The dried *L. chuanxiong* rhizome samples were crushed into powder and collected through a sieve with a pore. About one g of the powder was accurately weighed and put into a conical bottle with a stopper, and then 50 mL of 75% methanol solution was accurately added. Samples were ultrasonically extracted (power: 360 W, frequency: 40 kHz) for 30 min. The supernatant was taken and filtered by 0.22 $\mu$m microporous membrane, and the filtrate was taken to obtain the sample solution.

### Chromatographic conditions

Quantitative analysis was performed using an UltiMate 3000 UPLC system equipped with a DAD detector. The separation of the three active components was performed at 30 °C using an Agilent ZORBAX Eclipse Plus C18 column (2.1 × 50 mm, 1.8 $\mu$m). Use 0.03% phosphoric acid water (A)-methanol (B) as the gradient solvent system. The gradient elution conditions were as follows: 0~3 min, 15%~25% B; 3~5 min, 25%~28% B; 5~9 min, 28%~30% B; 10~16 min, 48%~50% B; 16~18 min, 50%~80% B. The flow rate and injection volume were 0.40 mL min$^{-1}$ and 1 $\mu$L, respectively. The detection wavelengths

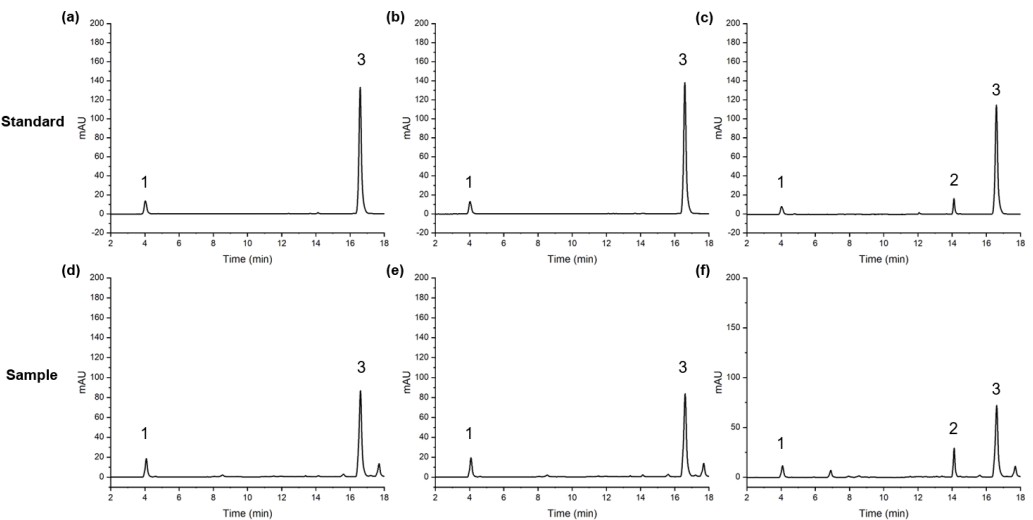

**Figure 2** The peaks of FA(1), SA(2) and Z-L(3) in standard and sample. **1**, FA; **2**, SA; **3**, Z-L; (A, D) 322 nm; (B, E) 328 nm; (C, F) 282 nm.

were 282 nm, 322 nm and 328 nm. The chromatographic peak separation results (Fig. 2) show that the chromatographic peak separation was good.

### Multispectral image processing

The spectral images of the five bands obtained from each flight mission were imported into Pix4D Mapper 4.5.6 to complete the geometric correction and radiation correction with image information from the corresponding radiation calibration panels. Then, the spectral images were spliced into orthographic mosaic reflectance images. ENVI 5.3 software was used to overlay the orthographic mosaics of each band into a multispectral image. After classifying the canopy of *L. chuanxiong* and the background of soil and weeds by the Support Vector Machine class function of ENVI, a mask was constructed for segmentation, and the multispectral image of *L. chuanxiong* was obtained (Fig. 3). By selecting the location of each field sampling area and editing the corresponding region of interest (ROI), the VIs of each ROI were extracted and calculated to obtain 23 characteristic variables (Table 1).

### VIs selection and models construction

The best model input variables were screened by Pearson correlation analysis between variables and the target values of FW, DW, FA, Z-L and SA in the corresponding samples. However, there may be a high degree of correlation and collinearity among variables. Screening only the characteristic variables with high Pearson correlation coefficient may lead to data redundancy and miss some effective information (*Xu et al., 2023*). Therefore, according to the order of the absolute value of the correlation coefficient between the variables and the target value from large to small, the characteristic variables with the correlation coefficient greater than 0.9 were eliminated sequentially. The remaining characteristic variables with significant correlation ($p < 0.05$) were used as model input variables to construct a prediction model corresponding to each target value.

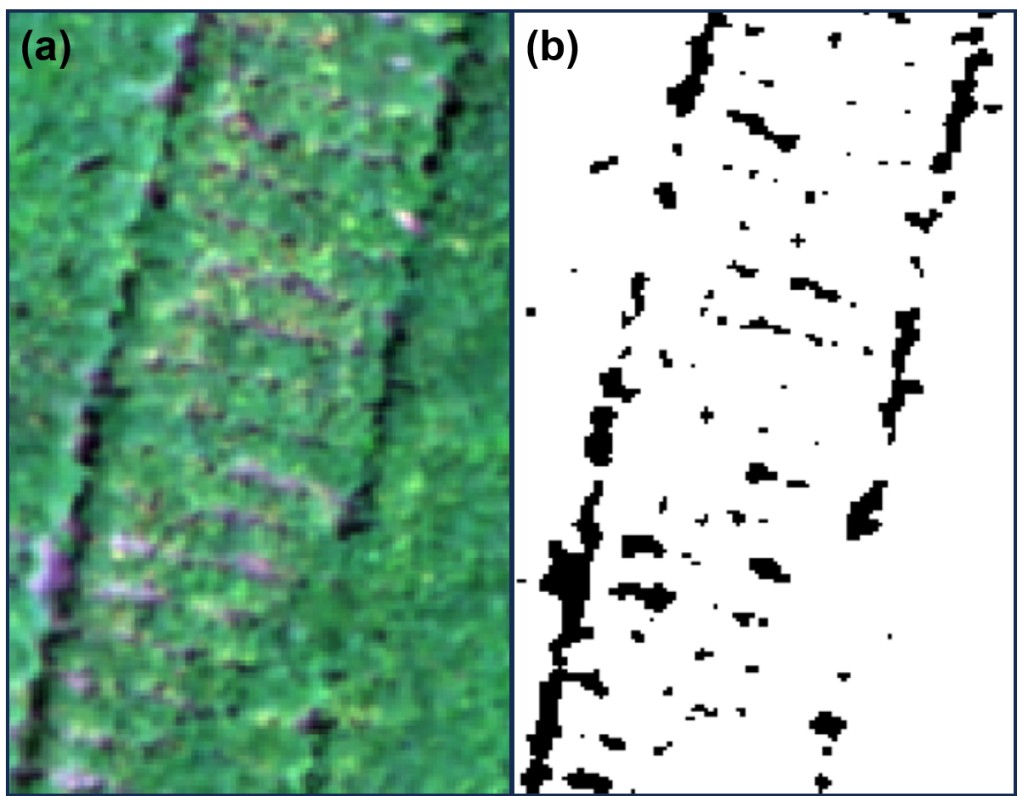

**Figure 3** **Mask extraction for separating the canopy of *L. chuanxiong* from the background.** (A) RGB image; (B) mask.

Model construction and related data processing were performed in Python 3.11. Four machine learning ensemble algorithms, Random Forest (RF), adaptive boosting (AdaBoost), Gradient Boosting Decision Tree (GBDT) and extreme gradient boosting (XGBoost), were used to construct regression prediction models of FW, DW, FA, Z-L and SA contents, respectively. The quality discriminant models were based on the standard set by the Pharmacopoeia of the People's Republic of China (2020) (ChP 2020) (*Chinese National Pharmacopoeia Commission, 2020*) that the FA content of chuanxiong rhizoma should not be less than 0.10% (1.0 mg g$^{-1}$). Considering that there may be a large difference between the sample sizes that meet the criteria or not, the Border line SMOTE was used to oversample the categories with a small sample size to obtain balanced data. Each data set was randomly divided into a training set and a testing set with a ratio of 7:3 for model building. In the process of model construction, each algorithm determined the optimal model parameters through grid search and five-fold cross validation.

Three indicators were used to evaluate the effectiveness and stability of the model. Coefficient of determination ($R^2$) is used to measure the fitting degree of the regression model, and the value was 0~1. The larger the value, the better the fitting degree of the model. Normalized root mean squared error (NRMSE) and mean absolute percentage error (MAPE) are indicators of the gap between the predicted value and the true value. The former

**Table 1** The characteristic variables used in this study.

| Variable | Name | Formula | Reference |
|---|---|---|---|
| R | Red | Band1 | \ |
| G | Green | Band2 | \ |
| B | Blue | Band3 | \ |
| RE | Red Edge | Band4 | \ |
| NIR | Near Infrared | Band5 | \ |
| r | Normalized Red | $R/(R+G+B)$ | *Tarbell & Reid (1991)* |
| g | Normalized Green | $G/(R+G+B)$ | *Tarbell & Reid (1991)* |
| b | Normalized Blue | $B/(R+G+B)$ | *Tarbell & Reid (1991)* |
| RBRI | Red Blue Ratio Index | $R/B$ | *Wu et al. (2023)* |
| GLI | Green Leaf Index | $(2G-R-B)/(2G+R+B)$ | *Louhaichi, Borman & Johnson (2001)* |
| MSRI | Modified Simple Ratio Index | $(NIR/RE-1)/\sqrt{NIR/RE+1}$ | *Chen (1996)* |
| OSAVI | Optimized Soil Adjusted Vegetation Index | $(NIR-R)/(NIR+R+0.16)$ | *Rondeaux, Steven & Baret (1996)* |
| RECI | Red Edge Chlorophyll Index | $NIR/RE-1$ | *Gitelson et al. (2005)* |
| RERDVI | Red Edge Re-nomalized Different Vegetation Index | $(NIR-RE)/\sqrt{(NIR+RE)}$ | *Roujean & Breon (1995)* |
| LCI | Leaf Chlorophyll Index | $(NIR-RE)/(NIR+R)$ | *Datt (1999)* |
| EVI | Enhanced Vegetation Index | $2.5(NIR-R)/(NIR+6R-7.5B+1)$ | *Justice et al. (1998)* |
| WI | Woebbecke Index | $(G-B)/(R-G)$ | *Woebbecke et al. (1995)* |
| GRVI | Green Ratio Vegetation Index | $NIR/G$ | *Buschmann & Nagel (1993)* |
| RVI | Ratio Vegetation Index | $NIR/R$ | *Jordan (1969)* |
| RERVI | Red Edge Ratio Vegetation Index | $NIR/RE$ | *Jasper, Reusch & Link (2009)* |
| BRVI | Blue Ratio Vegetation Index | $NIR/B$ | *Kandylakis & Karantzalos (2016)* |
| NGRVI | Normalized Green-Red Vegetation Index | $(G-R)/(G+R)$ | *Gitelson et al. (2002)* |
| NDVI | Normalized Difference Vegetation Index | $(NIR-R)/(NIR+R)$ | *Rouse et al. (1974)* |

is more sensitive to large errors because of the root mean square calculation, while the latter weights all errors equally. Both of them can compare the effects of models composed of different data sets. The performance of the model in the independent validation set was evaluated using NRMSE and MAPE. The calculation formula is as follows (*Costa et al., 2022*; *Zhu et al., 2023*):

$$R^2 = 1 - \sum_{i=1}^{n}(y_i - \hat{y}_i)^2 / \sum_{i=1}^{n}(y_i - \bar{y})^2 \tag{1}$$

$$NRMSE = (1/\bar{y})\sqrt{\sum_{i=1}^{n}(y_i - \hat{y}_i)^2/n} \cdot 100\% \tag{2}$$

$$MAPE = (1/n)\sum_{i=1}^{n}|y_i - \hat{y}_i|/y_i \cdot 100\% \tag{3}$$

where $n$ is the sample size of the input data, $y_i$ is the true target value, $\bar{y}$ is the average value of the corresponding true target value, and is $\hat{y}_i$ the predicted target value calculated by the corresponding regression model.

The performance of the quality discriminant model was evaluated using the following indicators: Accuracy is used to evaluate the overall discriminant effect of the discriminant model. Precision, recall and F1-score derived from the confusion matrix can explain the discriminant effect in more detail. The area under curve (AUC) is the area under the receiver operating characteristic (ROC) curve, and the value is 0~1. The larger the AUC, the better the discriminant effect of the model. The performance of the model in the independent validation set was evaluated using accuracy and AUC. The formula is as follows (*Zhang et al., 2022a*; *Zhang et al., 2022b*):

$$\text{Accuracy} = (TP + TN)/(TP + TN + FP + FN) \tag{4}$$

$$\text{Precision} = TP/(TP + FP) \tag{5}$$

$$\text{Recall} = TP/(TP + FN) \tag{6}$$

$$\text{F1} - \text{score} = 2 \times \text{Precision} \times \text{Recall}/(\text{Precision} + \text{Recall}) \tag{7}$$

where TP is the number of samples whose instances are positive and predicted to be positive, TN is the number of samples whose instances are negative and predicted to be negative, FN is the number of samples whose instances are positive and predicted to be negative, FP is the number of samples whose instances are negative and predicted to be positive.

## RESULTS

### Correlation analysis of VIs

According to the results of the correlation analysis (Fig. 4) combined with the feature screening method in VIs selection and models construction, the input characteristic variables of the FW prediction model in D1 were red edge (RE), optimized soil adjusted vegetation index (OSAVI), green (G), Woebbecke Index (WI), blue (B), modified simple ration index (MSRI), Green Leaf Index (GLI), normalized blue (b) and red blue ratio index (RBRI), while the input characteristic variables of the DW prediction model were RE, OSAVI, green (G), WI, B, normalized green-red vegetation index (NGRVI), radio vegetation index (RVI), GLI, b, red (R), and RBRI. In addition, red edge chlorophy II index (RECI), red edge re-normalized different vegetation index (RERDVI), LCI, enhanced vegetation index (EVI), RBRI, WI, G, green ration vegetation index (GRVI), B and r were the input characteristic variables of the FW and DW prediction models in D2.

The input characteristic variables of the FA content prediction model and the quality discriminant model in D1 were b, RBRI and B, while those in D2 were RE and red edge ratio vegetation index (RERVI). The input variables of the SA content prediction model in D1 were b, g, B, BRVI, R and r, while those in D2 were RE, GLI, g, RVI, and RERVI. The input variables of the Z-L content prediction model in D1 were G, RE, OSAVI, WI, RBRI, B and NGRVI, while those in D2 were G, B, RECI, NIR, RERDVI and WI.
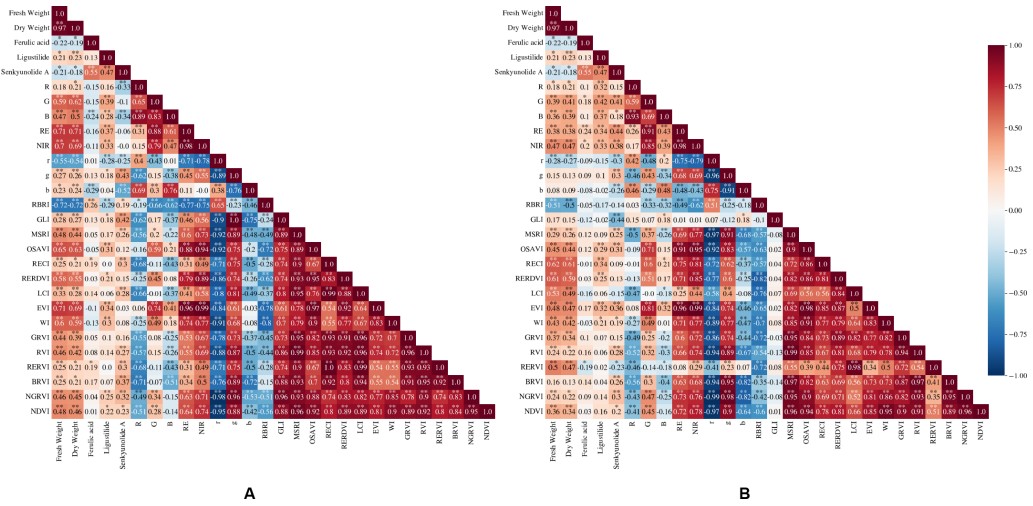

**Figure 4** Correlation coefficients between variables and the FW, DW and the content of FA, Z-L and SA in D1 and D2. (A) D1; (B) D2; *, $p < 0.05$; **, $p < 0.01$.

**Table 2  Evaluation of different FW prediction models.**

| Period | Model | Training set | | | Test set | | |
|---|---|---|---|---|---|---|---|
| | | $R^2$ | NRMSE | MAPE | $R^2$ | NRMSE | MAPE |
| D1 | XGBoost | 0.89 | 14.15% | 14.37% | 0.82 | 19.19% | 16.83% |
| | GBDT | 0.78 | 20.95% | 15.49% | 0.76 | 23.93% | 16.43% |
| | AdaBoost | 0.79 | 19.21% | 18.45% | 0.76 | 23.43% | 23.48% |
| | RF | 0.84 | 17.22% | 17.54% | 0.80 | 20.11% | 17.80% |
| D2 | XGBoost | 0.87 | 15.31% | 15.68% | 0.71 | 24.38% | 18.40% |
| | GBDT | 0.81 | 19.21% | 14.01% | 0.72 | 24.67% | 17.22% |
| | AdaBoost | 0.72 | 23.12% | 21.16% | 0.69 | 25.91% | 18.11% |
| | RF | 0.72 | 22.40% | 22.04% | 0.71 | 25.36% | 17.91% |

**Notes.**
D1 and D2 represent April 27 and May 8 respectively.

## Different prediction models
### Yield prediction models

The $R^2$ (Table 2) of the four algorithm models in the FW training set and the test set data in both periods could reach more than 0.68. Overall, the model with the best prediction accuracy in both periods was XGBoost, the $R^2$ of the training set was close to 0.90, and the NRMSE and MAPE were less than 20.00%. The XGBoost over-fitting degree in D1 was lower than that in D2, whose $R^2$ on the testing set was 19.54% lower than that on the training set.

The difference between the DW prediction models (Table 3) was similar to that of FW, and the best prediction model was the XGBoost model in D1. The $R^2$ on the training and test sets were both greater than 0.80, and the NRMSE and MAPE were within the range from 19.00% to 21.00%. In the model in D2, the XGBoost model performed best in the

**Table 3** Evaluation of different DW prediction models.

| Period | Model | Training set | | | Test set | | |
|--------|-------|------|-------|------|------|-------|------|
| | | R² | NRMSE | MAPE | R² | NRMSE | MAPE |
| D1 | XGBoost | 0.82 | 20.04% | 20.85% | 0.81 | 20.94% | 19.02% |
| | GBDT | 0.77 | 23.40% | 17.52% | 0.76 | 25.16% | 20.54% |
| | AdaBoost | 0.79 | 21.54% | 25.03% | 0.78 | 21.94% | 25.44% |
| | RF | 0.80 | 20.82% | 22.96% | 0.79 | 21.88% | 22.52% |
| D2 | XGBoost | 0.82 | 19.77% | 22.49% | 0.70 | 27.15% | 22.15% |
| | GBDT | 0.81 | 21.07% | 14.92% | 0.74 | 24.71% | 22.18% |
| | AdaBoost | 0.71 | 25.30% | 25.78% | 0.68 | 28.53% | 22.36% |
| | RF | 0.70 | 26.71% | 22.27% | 0.69 | 24.14% | 30.40% |

**Notes.**
D1 and D2 represent April 27 and May 8 respectively.

**Table 4** Evaluation in D1 prediction model of different active component contents.

| Period | Model | Training set | | | Test set | | |
|--------|-------|------|-------|------|------|-------|------|
| | | R² | NRMSE | MAPE | R² | NRMSE | MAPE |
| FA | XGBoost | 0.53 | 18.88% | 15.86% | 0.39 | 20.95% | 18.01% |
| | GBDT | 0.81 | 12.31% | 7.94% | 0.44 | 20.30% | 17.27% |
| | AdaBoost | 0.58 | 17.64% | 16.38% | 0.43 | 20.23% | 17.75% |
| | RF | 0.45 | 20.63% | 18.75% | 0.12 | 24.56% | 20.04% |
| Z-L | XGBoost | 0.34 | 19.01% | 17.69% | 0.22 | 23.59% | 25.12% |
| | GBDT | 0.59 | 15.23% | 13.49% | 0.21 | 23.73% | 23.59% |
| | AdaBoost | 0.49 | 17.52% | 17.31% | 0.25 | 19.75% | 21.13% |
| | RF | 0.79 | 10.70% | 9.81% | 0.28 | 22.74% | 23.42% |
| SA | XGBoost | 0.81 | 15.01% | 13.34% | 0.57 | 28.64% | 27.74% |
| | GBDT | 0.85 | 13.36% | 12.68% | 0.68 | 24.85% | 25.27% |
| | AdaBoost | 0.74 | 17.72% | 19.21% | 0.70 | 22.25% | 25.80% |
| | RF | 0.89 | 11.24% | 10.63% | 0.63 | 27.00% | 26.09% |

**Notes.**
FA, Z-L and SA represent ferulic acid, Z-ligustilide and senkyunolide A, respectively.

training set with an $R^2$ of 0.82, but the over-fitting situation was still higher than other algorithm models.

### Active component contents prediction models

The effects of FA, Z-L and SA content prediction models in D1 are shown in Table 4. Among the FA content prediction models in D1, the training set $R^2$ of GBDT was the largest, but the $R^2$ of the test set decreases greatly, up to 45.68%, while the AdaBoost model has a small degree of over-fitting while ensuring a low error. The degree of over-fitting of the Z-L content prediction models was high, and the AdaBoost had the lowest degree of over-fitting that the difference between the $R^2$ values of the training set and the test set was 0.24. The $R^2$ of all SA content prediction models in D1 performed well and the degree of over-fitting was low. Among them, the RF training set had the highest $R^2$ of 0.89, while the $R^2$ of the AdaBoost test set was higher and the degree of over-fitting was lower.

**Table 5  Evaluation in D2 prediction model of different active component contents.**

| Period | Model | Training set | | | Test set | | |
|---|---|---|---|---|---|---|---|
| | | R² | NRMSE | MAPE | R² | NRMSE | MAPE |
| FA | XGBoost | 0.68 | 15.75% | 13.77% | 0.35 | 20.59% | 17.95% |
| | GBDT | 0.45 | 20.74% | 19.08% | 0.24 | 20.18% | 19.60% |
| | AdaBoost | 0.35 | 23.76% | 19.34% | 0.30 | 20.15% | 16.76% |
| | RF | 0.78 | 13.00% | 11.40% | 0.33 | 20.78% | 18.74% |
| Z-L | XGBoost | 0.69 | 13.00% | 12.45% | 0.34 | 21.97% | 24.38% |
| | GBDT | 0.63 | 14.25% | 13.43% | 0.31 | 22.32% | 23.67% |
| | AdaBoost | 0.51 | 17.25% | 15.39% | 0.39 | 18.06% | 20.70% |
| | RF | 0.82 | 10.62% | 10.45% | 0.32 | 19.62% | 16.93% |
| SA | XGBoost | 0.86 | 13.38% | 13.10% | 0.75 | 21.17% | 21.49% |
| | GBDT | 0.90 | 11.54% | 6.56% | 0.76 | 21.59% | 17.56% |
| | AdaBoost | 0.74 | 19.06% | 17.95% | 0.74 | 20.58% | 17.97% |
| | RF | 0.89 | 11.68% | 10.81% | 0.67 | 24.44% | 23.76% |

**Notes.**
FA, Z-L and SA represent ferulic acid, Z-ligustilide and senkyunolide A, respectively.

The effect of the three active component content prediction models in D2 is shown in Table 5. The FA and Z-L content prediction models of D2 had higher degree of over-fitting, except for AdaBoost. However, each training set $R^2$ of AdaBoost was lower, 0.35 and 0.51, respectively. The $R^2$ of the SA content prediction model in D2 was higher, and GBDT was the best, whose $R^2$ of the training set and the test set were 0.90 and 0.76, respectively. While the AdaBoost, with training set $R^2$ of 0.74 for both training and test sets, had the lowest degree of over-fitting.

### Chuanxiong rhizoma quality discriminant models

ChP 2020 stipulates that the FA content in chuanxiong rhizoma should not be less than 0.10% (1.00 mg g$^{-1}$) (*Chinese National Pharmacopoeia Commission, 2020*). Therefore, we constructed quality discriminant models (Table 6) to evaluate whether the FA content reached the standard. The accuracy of XGBoost and GBDT models in D1 reached 0.9020, and the AUC was 0.9031 and 0.9095, respectively. The Accuracy of XGBoost, AdaBoost and RF constructed by D2 data was better and all of them were 0.8431. Among them, the AUC of RF was larger so the discriminative efficiency was the best.

## Independent validation
### Independent validation of yield prediction models

Since D2 and D3 are close to the harvesting periods of Shiyang Town and Aoping Town respectively, the VIs of D3 were used as the characteristic variable to be input according to the requirements of the yield prediction models in D2 to evaluate the performance of models on the independent validation set. As shown in Fig. 5, the relationship between the predicted value and the actual value is obtained. When applied to the independent validation set, each model underestimates FW and DW. Among them, RF was the best among the FW prediction models with NRMSE of 23.76%, and MAPE of 14.75%, while

**Table 6  Evaluation of different discriminant models.**

| Period | Model | Whether achieved | Accuracy | Precision | Recall | F1 Score | AUC |
|---|---|---|---|---|---|---|---|
| D1 | XGBoost | No | 0.9020 | 0.8571 | 0.9600 | 0.9057 | 0.9031 |
| | | Yes | | 0.9565 | 0.8462 | 0.8900 | |
| | GBDT | No | 0.9020 | 0.8333 | 0.9524 | 0.8889 | 0.9095 |
| | | Yes | | 0.9630 | 0.8667 | 0.9123 | |
| | AdaBoost | No | 0.8235 | 0.7308 | 0.9048 | 0.8085 | 0.8357 |
| | | Yes | | 0.9200 | 0.7667 | 0.8364 | |
| | RF | No | 0.8431 | 0.7778 | 0.9130 | 0.8400 | 0.8494 |
| | | Yes | | 0.9167 | 0.7857 | 0.8462 | |
| D2 | XGBoost | No | 0.8431 | 0.8387 | 0.8966 | 0.8667 | 0.8346 |
| | | Yes | | 0.8500 | 0.7727 | 0.8095 | |
| | GBDT | No | 0.8235 | 0.8636 | 0.7600 | 0.8085 | 0.8223 |
| | | Yes | | 0.7931 | 0.8846 | 0.8364 | |
| | AdaBoost | No | 0.8431 | 0.8276 | 0.8889 | 0.8571 | 0.8403 |
| | | Yes | | 0.8636 | 0.7917 | 0.8261 | |
| | RF | No | 0.8431 | 0.7917 | 0.8636 | 0.8261 | 0.8456 |
| | | Yes | | 0.8889 | 0.8276 | 0.8571 | |

**Notes.**
D1 and D2 represent April 27 and May 8 respectively.

AdaBoost was the best among the DW prediction models with NRMSE of 34.60% and MAPE of 21.73%.

***Independent validation of active component content prediction models***
The prediction models of FA, Z-L and SA content in D2 were applied to predict the content of active components in the independent validation set, and the results were shown in Fig. 6. AdaBoost performs best in predicting FA content. Compared to the other models, the RF has a more concentrated distribution of the predicted values and smaller relative errors. The predicted Z-L content of each model was generally lower than the true values, among which the RF's predicted values were the most concentrated with NRMSE of 34.35% and MAPE of 23.40%. The predicted SA content of each model was generally lower than the true values. Among them, XGBoost was the best at predicting with NRMSE of 45.26% and MAPE of 25.48%.

***Independent validation of chuanxiong rhizoma quality discriminant models***
To verify the effect of the constructed quality discriminant model based on the FA content standard of chuanxiong rhizoma stipulated in ChP 2020, the corresponding characteristic variables were entered into the quality discriminant model of D2, and the discriminant results were shown in Fig. 7. The best performing models were XGBoost and RF, with accuracy and AUC of 0.7083 and 0.7214, respectively, which had effective discriminative ability.

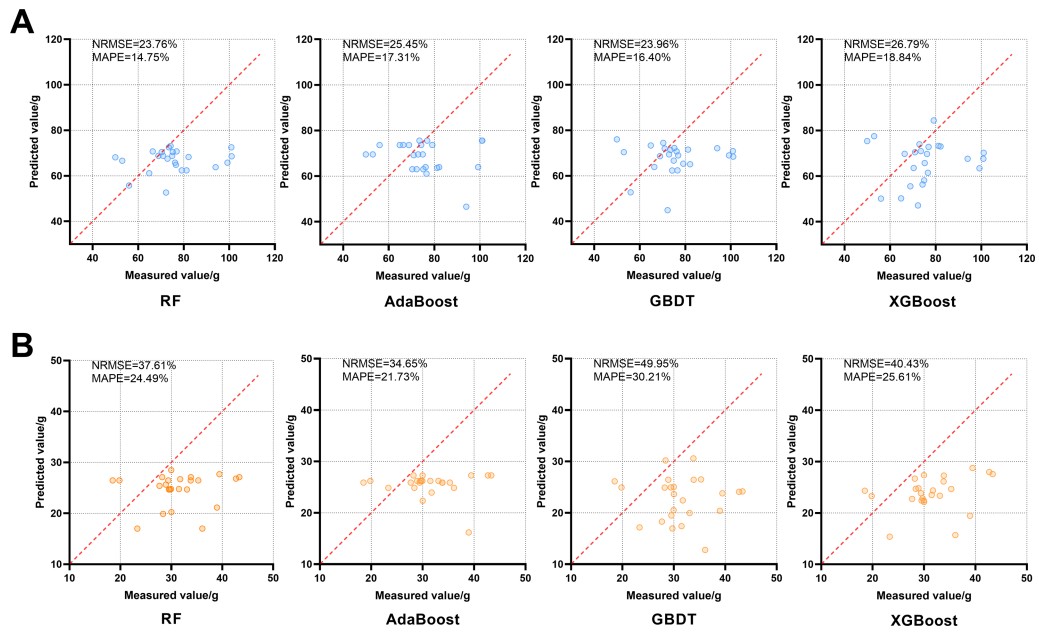

**Figure 5 Effects of four FW and DW prediction models, RF, AdaBoost, GBDT and XGBoost, on the independent validation set.** The red dotted line in each image is a 1:1 line. (A) FW; (B) DW.

## DISCUSSION

### Different sensitivity of variables

The four machine learning algorithms selected in this study could all use the variables selected in the two periods to train effective prediction models for predicting the yield and quality of *L. chuanxiong*. There were differences in the input variables of the yield prediction models in the two periods (Fig. 4), but the most advanced variables (RE, OSAVI, RECI, RERDVI, *etc.*) were related to the red edge and near-infrared bands, which indicated that the variables based on the reflectance of the red edge and near-infrared bands could better reflect the FW and DW of rhizome of *L. chuanxiong*. This is consistent with the results obtained by *Mutanga, Adam & Cho (2012)* who constructed a wetland biomass estimation model. In addition, the input variables of the three active component content prediction models selected in the two periods were quite different (Fig. 4). For example, the top input variables of the FA and SA content prediction models in the D1 variables (b, B, BRVI, RBRI, *etc.*) are mainly related to the blue band, while the D2 variables (RE, RERVI, *etc.*) are mainly related to the red edge band. The difference in this short-term interval may be due to the fact that April to May is the period when *L. chuanxiong* changes from the end of the vigorous growth period of the aboveground part to the rapid expansion of the underground part. The growth rate of the new leaves on the aboveground part gradually stops, the old leaves gradually wither and the leaf water content changes greatly (*Zhao, Fu & Fan, 2008*), resulting in a certain change in the canopy nutrient structure, which affected the reflectance of each band. However, the sensitivity of *L. chuanxiong* to the reflectance of

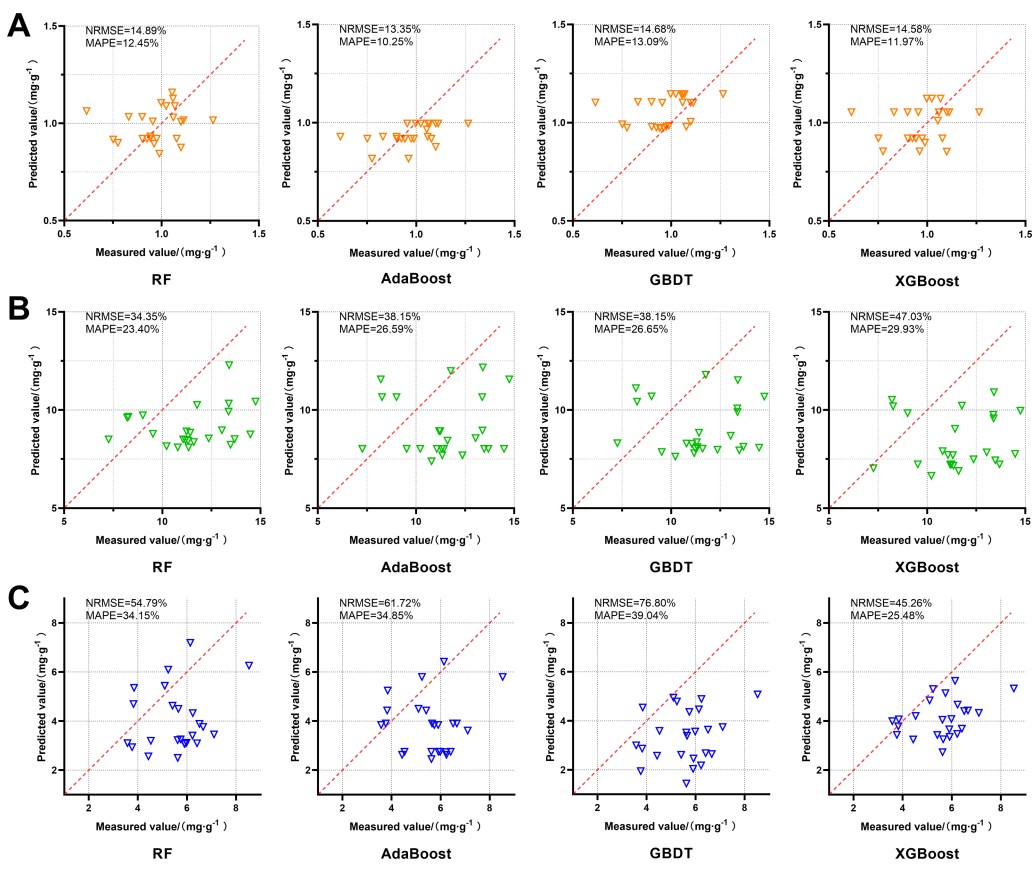

**Figure 6** **Effects of four FA, Z-L and SA prediction models, RF, AdaBoost, GBDT and XGBoost, on the independent validation set.** The red dotted line in each image is a 1:1 line. (A) FA; (B) Z-L; (C) SA.

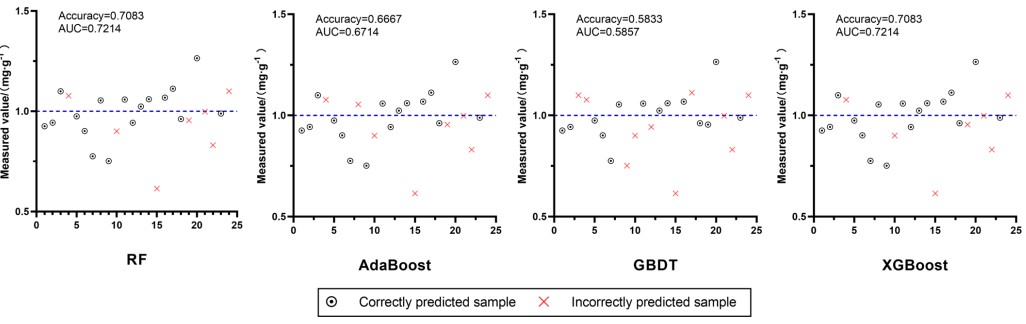

**Figure 7** **Effects of four quality discriminant models, RF, AdaBoost, GBDT and XGBoost, on independent validation set.** The blue dotted line in each image is Measured value = 1.0 mg g$^{-1}$.

different bands in this study is different from the results of *Fan et al. (2024)*. They found that although there were differences in the variables sensitive to potato yield at different growth stages, the blue band was always closely related to yield, and the near-infrared band

could reflect the difference in growth between plants. This may be due to the difference in different crops with the different sensitivity of variables.

## Different performances of models

The XGBoost performed best in predicting the yield data of Shiyang Town in both periods, and the degree of over-fitting was low (Tables 2 and 3). Similarly, XGBoost performed better than other models in studies such as *Shahi et al. (2023)*, which used UAV images of five growth stages of peanut to predict yield, and *Wang et al. (2023b)*, which used UAV multispectral data to evaluate the above-ground biomass of camphor trees. However, in this study, RF and AdaBoost had the best predictive effect on FW and DW in the independent validation set, and XGBoost performed slightly worse (Fig. 5). This may be because the RF and AdaBoost have fewer hyperparameters, and the model is simpler and has better generalization ability than XGBoost. In addition, the performance of the all models on the independent validation set shows that the predicted value is generally lower than the true value. This is contrary to the results of *Liu et al. (2019)*, who applied the winter oilseed rape biomass prediction models to the independent validation data set of another year's and they believe that this is due to the large climatic differences between the experimental sites in different years. The collection period of the independent validation set of this study is similar to that of the data set collected to construct the models, therefore, the main difference in the sets reflected in the spatial difference. The low predicted values may be mainly related to the differences between the soil (*Peng et al., 2021*) and ecological environmental factors (*Yin et al., 2012*) in the two locations.

In the performance of predicting the content of the three active components, AdaBoost had a lower degree of over-fitting in D1 and D2. This is similar to the results of *Yoon et al. (2023)* using different algorithms combined with hyperspectral prediction of mustard metabolites. In that study, AdaBoost predicts secondary metabolites such as phenolics and flavonoids better than XGBoost and LightGBM. In the performance of the independent validation set (Fig. 6), the best models for predicting FA and Z-L content were AdaBoost (NRMSE = 13.35%, MAPE = 10.25%) and RF (NRMSE = 34.35%, MAP = 23.40%), respectively, and they had lower relative errors. The best model for SA content prediction is XGBoost, but there is still a large relative error (NRMSE = 45.26%, MAPE = 25.48%). The results also reflect the advantages of the model with fewer hyperparameters in universality, and show that different models have different performances in predicting the content of different active components.

In addition, we also constructed the quality discriminant models of chuanxiong rhizoma according to the standard that the FA content should not be less than 0.10% (1.00 mg $g^{-1}$), which is stipulated in ChP 2020. The accuracy and AUC of the models in each period (Table 6) were greater than 0.8 in the test set. Among them, the Accuracy and AUC of XGBoost and GBDT models in D1 could reach more than 0.9, and the those of XGBoost, AdaBoost and RF models in D2 could reach more than 0.84. The models in D2 were applied to the independent validation set, and the effects of XGBoost and RF performed best that the Accuracy and AUC were both 0.7083 and 0.7214, respectively (Fig. 7). Although the performance of the models decreased to some extent in the independent validation set,

they still had effective discriminatory ability, indicating that it was feasible to use VIs to construct a discriminant model based on FA content to evaluate the quality of chuanxiong rhizoma.

## CONCLUSIONS

In this study, the models for predicting the fresh weight, dry weight of *L. chuanxiong* rhizome and the contents of ferulic acid, Z-ligustilide and senkyunolide A in chuanxiong rhizoma were constructed on the basis of UAV multispectral images. And the quality discriminant models of chuanxiong rhizoma based on the standard that the FA content should not be less than 0.10% (1.00 mg g$^{-1}$) stipulated in the Pharmacopoeia of the People's Republic of China (2020) were constructed for the first time. In addition, the verification of the models on the independent validation set was completed, and the results proved that each prediction model had a certain degree of universality. According to the results of this study, we believe that UAV multispectral has great potential in non-destructively predicting the yield and quality of medicinal plants before harvesting. It can further provide a reasonable reference for the appropriate specification and classification of medicinal materials in production, as well as the management methods of processing, transporting and storing, so as to ensure the quality of medicinal materials and make more effective use of the resources.

### Funding

This study was funded by the Science and Technology Department of Sichuan Province (No. 2021YFS0045) and Chengdu University of Traditional Chinese Medicine (No. XJ2023001601). The funders had no role in study design, data collection and analysis, decision to publish, or preparation of the manuscript.

### Grant Disclosures

The following grant information was disclosed by the authors:
Science and Technology Department of Sichuan Province: No. 2021YFS0045.
Chengdu University of Traditional Chinese Medicine: No. XJ2023001601.

### Competing Interests

The authors declare there are no competing interests.

### Author Contributions

- Yun-Fan Li conceived and designed the experiments, performed the experiments, analyzed the data, prepared figures and/or tables, authored or reviewed drafts of the article, and approved the final draft.
- Chen Wu performed the experiments, analyzed the data, prepared figures and/or tables, authored or reviewed drafts of the article, and approved the final draft.
- Hong-Mei Jia performed the experiments, analyzed the data, prepared figures and/or tables, authored or reviewed drafts of the article, and approved the final draft.

- Xi Chen performed the experiments, prepared figures and/or tables, and approved the final draft.
- Jin-Niu Xing performed the experiments, authored or reviewed drafts of the article, and approved the final draft.
- Wei-Ping Gao performed the experiments, authored or reviewed drafts of the article, and approved the final draft.
- Zhu-Yun Yan conceived and designed the experiments, authored or reviewed drafts of the article, and approved the final draft.

## Data Availability

The raw yield and quality-related data are available in the Supplemental Files.

## Supplemental Information

Supplemental information for this article can be found online at http://dx.doi.org/10.7717/peerj.19264#supplemental-information.

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
