# Peer review of "Prediction of yield and quality in medicinal plant Ligusticum chuanxiong Hort. using uncrewed aerial vehicle multispectral measurement"

_PeerJ, doi:10.7717/peerj.19264_

## Round 0.1 · original submission · Major Revisions

Dear authors, I ask you to carefully answer the reviewers' fundamental questions and improve this manuscript in accordance with their comments.

·

Basic reporting

(1)The English must be improved to more clearly express the research. Spoken English and Chinglish are overly prominent.I suggest you have a colleague who is proficient in English and familiar with the subject matter review your manuscript, or contact a professional editing service.
(2)The introduction does not sufficiently highlight the research gap or novelty of the study. Please provide an explanation of the study's challenges and clarify how the suggested approach tackles and improves these challenges. Although my expectations are modest, I hope you can try your best to reorganize the introduction to emphasize the challenges and potential solutions of UAV remote sensing technology in monitoring the yield and quality of medicinal plants. Additionally, please identify the research entry point, the scientific problems this study aims to address, and the research gap it seeks to fill.
(3)The third paragraph lacks a clear connection to the first two paragraphs.It can be simplified and repositioned as the opening paragraph, with the addition of research gap before introducing the study's objectives.
(4)The sections are well-organized and easy to follow. No major issues were noted.
(5)The manuscript does not provide raw UAV data, which is essential for transparency and reproducibility. Please include the raw UAV dataset and provide clear instructions on how to access it.

Experimental design

(1)Detailed experimental design information for the study is unavailable. Please provide an outline of the similarities and differences in experimental design between the two sites.
(2)Lines111-115: This section does not provide a description of the study area.
(3)Only 88 samples(125 * 0.7=88) were used to train the model, which is a sample size too small for an ensemble learning model.Moreover, such ensemble learning models are often challenging to implement in agricultural practice.I encourage the author to carefully reconsider the necessity of using machine learning models, rather than adopting them solely for the purpose of publication.This is, of course, merely a suggestion and does not require modification.My concern is that the author may have inadvertently fallen into a common research trap.

Validity of the findings

Multispectral (MS) sensors measure reflectance across specific spectral bands beyond visible light, enabling advanced vegetation indices and crop assessments.Machine learning (ML) is defined as a subset of artificial intelligence (AI) involving algorithms that learn from data to identify patterns and make predictions. The models applied here include supervised learning techniques designed to correlate image-derived features with measured medicinal plants parameters.In this study, UAV multispectral remote sensing technology and ML were utilized for monitoring the yield and quality of medicinal plants, thereby expanding the application scope of UAV remote sensing technology and offering a novel tool for monitoring the growth of medicinal plants.
(1)The FA can be considered as involving two processes: a prediction process (FA content prediction model) and a classification process (quality discriminant model). What is the distinction between these two processes?
(2)The process for converting consecutive FA values into classes should be explained. a conversion table should be included in the supplementary material, even if a classification standard is already available.
(3)The discussion and conclusion sections are well-structured and provide adequate interpretation of the results. If the introduction is refined to this level, it will enhance the scientific rigor of the study.

Additional comments

(1)Missing keywords.
(2)Line58: 'UAV' should be expanded to its full form.
(3)Line92: “In the previous study, our team used UAV multispectral to construct a detection model of water and fertilizer deficit in the growth period of L. chuanxiong”,Please check the English writing.
(4)Lines 126-127: What is the significance and rationale for selecting these two periods?
(5)Line179-181: What method is used to separate soil and background using a mask? How is the segmentation performed, and what is its accuracy?
(6)Line214-216: Why is R2 not used to evaluate the quality of the models for Independent validation(Figure 4 and 5), given that NRMSE and MAPE are statistically consistent in assessing significance?
(7)Figure 1 should be Sichuan, not Sicuan.

·

Basic reporting

Comments
The manuscript explores the important medicinal plant Ligusticum chuanxiong Hort. and aims to predict its yield and quality using uncrewed aerial vehicle (UAV) multispectral data. This topic is of considerable interest to both agricultural science and pharmacology, especially given the plant's medicinal properties. However, there are a few critical issues that should be addressed to improve the manuscript's suitability for this journal.
Abbreviation List: The abbreviation list should be placed before the introduction to make it easier for readers to reference as they read through the paper.
Active Compounds: While the manuscript focuses on ferulic acid, Z-ligustilide, and senkyunolide A, it overlooks the analysis of chuanxiongzine. Given that chuanxiongzine is also an effective active compound in this plant, its inclusion would strengthen the paper’s findings and provide a more complete picture.
Introduction Section: In the introduction, it would be beneficial to compare the production rate of the chuanxiong rhizome before and after the application of UAV technology. A comparative study would offer useful insights into the effectiveness of the UAV approach in improving yield and quality.
Figure Quality: The quality of Figure 2 is quite poor. It would be helpful to separate the sample and standard chromatograms for better clarity and visual presentation.
While the manuscript shows great potential, substantial revisions are necessary to address the concerns outlined above. I recommend a minor revision to enhance the manuscript's clarity and overall impact.

Experimental design

Comments
The manuscript explores the important medicinal plant Ligusticum chuanxiong Hort. and aims to predict its yield and quality using uncrewed aerial vehicle (UAV) multispectral data. This topic is of considerable interest to both agricultural science and pharmacology, especially given the plant's medicinal properties. However, there are a few critical issues that should be addressed to improve the manuscript's suitability for this journal.
Abbreviation List: The abbreviation list should be placed before the introduction to make it easier for readers to reference as they read through the paper.
Active Compounds: While the manuscript focuses on ferulic acid, Z-ligustilide, and senkyunolide A, it overlooks the analysis of chuanxiongzine. Given that chuanxiongzine is also an effective active compound in this plant, its inclusion would strengthen the paper’s findings and provide a more complete picture.
Introduction Section: In the introduction, it would be beneficial to compare the production rate of the chuanxiong rhizome before and after the application of UAV technology. A comparative study would offer useful insights into the effectiveness of the UAV approach in improving yield and quality.
Figure Quality: The quality of Figure 2 is quite poor. It would be helpful to separate the sample and standard chromatograms for better clarity and visual presentation.
While the manuscript shows great potential, substantial revisions are necessary to address the concerns outlined above. I recommend a minor revision to enhance the manuscript's clarity and overall impact.

Validity of the findings

The manuscript explores the important medicinal plant Ligusticum chuanxiong Hort. and aims to predict its yield and quality using uncrewed aerial vehicle (UAV) multispectral data. This topic is of considerable interest to both agricultural science and pharmacology, especially given the plant's medicinal properties. However, there are a few critical issues that should be addressed to improve the manuscript's suitability for this journal

---

## Round 0.2 · Major Revisions

Dear authors, I ask you to very carefully correct the manuscript in accordance with the reviewer's comments.

·

Basic reporting

Although the author has provided reasonable explanations for most of the comments, some problems still remain.
1、Line30: “The results indicated that the models had a considerable effect”,Incomprehensible expression.
2、Lines 80 - 92: It is unclear what the author is trying to convey. This section of text has no logical connection with the preceding or following content. Please read my previous review comments carefully and make additional revisions regarding the research gaps and scientific issues in using UAV multispectral imagery for predicting yield and quality of of chuanxiong rhizoma.
Does the author think that simply adjusting the order of sentences constitutes a revision?
3、In "VIs selection and models construction" section, please add references.
4、Please provide the R² values in Figures 5 and 6.

Experimental design

Lines 109 - 112: Is this part closely related to the study area? Please provide a description of the study area.

Validity of the findings

No

Additional comments

No

·

Basic reporting

The authors have adequately addressed all the queries. I recommend moving forward with the publication from my end.

Experimental design

Good and well design

Validity of the findings

Sufficient for publication

Additional comments

I have no any additional comments

---

## Round 0.3 · accepted · Accept

Dear authors, I am pleased to inform you that your article has been accepted for publication.

The Section Editor noted:

> I think the title (and the same phrase in the abstract and maybe elsewhere) should be "uncrewed aerial vehicle multispectral measurement" rather than "uncrewed aerial vehicle multispectral". +++ line 22 should "experiential" be "experimental"?

·

Basic reporting

The author has made the necessary revisions as per the requirements, and no further suggestions for improvement are necessary. Additionally, I encourage researchers to carefully consider whether applying complex machine learning algorithms to very small sample sizes is truly warranted (which also explains the negative R2 in the validation set). While machine learning is a popular field, its applicability should be evaluated beforehand, rather than blindly following trends. The authors should focus more on addressing the core issues within the field. I look forward to seeing more outstanding research from the authors in the future.

Experimental design

No

Validity of the findings

No

Additional comments

No